# Occupations and Risk of Head and Neck Cancers: A Case–Control Study in Tanzania

**DOI:** 10.3390/ijerph22111643

**Published:** 2025-10-29

**Authors:** Luco Patson Mwelange, Israel Paul Nyarubeli, Gloria Sakwari, Simon Henry Mamuya, Bente Elisabeth Moen

**Affiliations:** 1Department of Environmental and Occupational Health, Muhimbili University of Health and Allied Sciences, Dar es Salaam P.O. Box 65015, Tanzania; mwelange@gmail.com (L.P.M.); israel.nyarubeli@uib.no (I.P.N.); gsakwari@gmail.com (G.S.); mamuyasimon2@gmail.com (S.H.M.); 2Department of Global Public Health and Primary Care, Centre for International Health, University of Bergen, 5020 Bergen, Norway

**Keywords:** agriculture, cancer, head and neck cancer, occupation, Tanzania

## Abstract

Cancer is a major global health concern. Head and neck cancers are the sixth most prevalent type of cancer globally; it has been suggested that these cancers can be caused due to pesticide exposure during agricultural activities. In this study, we aimed to investigate whether agricultural labor is associated with an increased risk of head and neck cancers. A case–control study was performed in Ocean Road Cancer Institute in Tanzania: a national, specialized cancer hospital. A total of 298 head and neck cases and 305 controls were included. Occupational history and information about lifestyle factors and diet were obtained by interview. Using logistic regression analyses and adjusting for lifestyle and diet, an increased risk of head and neck cancer was found among workers with a history of agricultural work, with an odds ratio of 2.6 and a 95% confidence interval of 1.60–4.37. When including only non-smokers and non-alcohol users (*n* = 363), a similar estimate was found. Participants with over 10 years (*n* = 481) of agricultural work experience, after adjusting for lifestyle and diet, exhibited an odds ratio of 5.1, with a 95% confidence interval of 2.56–9.94. Our findings indicate that agricultural work is associated with the risk of head and neck cancer. Carcinogens in agriculture should be examined in future studies.

## 1. Introduction

In the twenty-first century, cancer represents a major social, public health, and economic challenge, accounting for approximately one in six deaths globally and one in four deaths from noncommunicable diseases (NCDs) [1]. One of the leading contributors to cancer-related mortality is head and neck cancer (HNC), which ranks as the sixth most common type of cancer globally. Nearly 82% of HNC-related deaths and 67% of new cases occur in developing countries such as Africa [1,2]. Data from the literature indicate that the recurrence prevalence of HNC in Africa is 15.4% [2], reflecting a growing disease burden, with an estimated annual incidence of 550,000–690,000 cases and an annual mortality of 300,000–450,000 cases [1,3]. In Tanzania, hospital records show that 7% of all cancer cases involve the head and neck, a figure projected to double by 2030 [3]. 

Cancer has multifactorial causes [4]. Modifiable risk factors include smoking, alcohol consumption, lifestyle choices, air pollution, and exposure to occupational and environmental carcinogens. In contrast, non-modifiable factors include genetics and aging [5]. Among the modifiable factors, alcohol consumption and cigarette smoking are well-established lifestyle contributors to various cancers, including HNC [6,7]. However, other modifiable risk factors, such as occupational exposure to carcinogens, have received less attention, particularly in developing countries [8].

Reducing occupational exposure or eliminating hazardous substances from the workplace can significantly lower the risk of occupational cancers [9]. Based on prior findings, the estimated occupation-attributable percentage for total cancer diagnoses typically varied between 2% and 8% (men, 3–14%; women, 1–2%) [10]. For instance, one study found that the risk of HNC increases with longer employment duration in industrial and agricultural settings [11]. However, this study was conducted in Iran, where the economy is better than in Tanzania. This makes it likely that also working conditions are better in Iran than in Tanzania, and the situation in the two countries might not be totally comparable. The scarcity of cancer research from developing countries has been highlighted in a previous review [9]. Nonetheless, evidence suggests that industrial and agricultural workplaces in developing countries often involve high exposure rates to carcinogens. For example, the chemicals formaldehyde, trichloroethylene, and cadmium are used in industries, and tetramethrin, chlorothalonil, malathion and glyphosate are used in agriculture [8]. The enforcement of safety regulations is weak in many developing countries, and the workers are not likely to be protected against the exposure to these chemicals [8]. In Tanzania, the use of carcinogenic chemicals in agriculture has been documented [12]. Furthermore, a previous study using secondary data found that over 50% of HNC patients had current or past involvement in agricultural work [3].

Based on the documented presence of carcinogens in agriculture and findings from prior research, we hypothesize that agricultural work may be a significant risk factor for HNC. Investigating occupational exposure to carcinogens in the agricultural sector within the Tanzanian context could provide valuable insights for policymakers, enabling targeted interventions to mitigate these risks. Given the multifactorial nature of HNC, this study aims to examine whether agricultural work is associated with an increased risk of developing HNC in Tanzania, while also accounting for lifestyle factors and dietary factors among affected patients.

## 2. Materials and Methods

### 2.1. Study Design and Setting

This case–control study was conducted in Tanzania, at the Ocean Road Cancer Institute (ORCI) in Dar es Salaam, from February 2023 to January 2024. The ORCI is a public national specialized hospital for cancer treatment in Tanzania. The ORCI was established in June 1996 by the Ocean Road Cancer Institute Act No. 192 of 1996. The hospital provides cancer services, such as screening, radiotherapy, chemotherapy, PET-CT scans, and palliative care. It is also the national referral hospital for cancer services and serves patients from around Tanzania and nearby countries.

### 2.2. Study Population

The study population consisted of unmatched cases and controls. The cases were HNC patients attending the treatment clinic at ORCI. Eligible cases were all individuals aged 18 years and older, currently undergoing treatment for HNC at ORCI. 

Community-based controls were selected to represent the exposure pattern of the general population from which the cases originated. A random walk sampling method was employed, with researchers accompanied by a community health worker to facilitate navigation and engagement. Streets located on the outskirts of Dar es Salaam were randomly selected, as these areas are characterized by a mix of industrial and agricultural activities. This approach ensured that the control group was selected from a population with comparable socioeconomic and environmental exposures to those of the cases. Within each selected street, the sampling began at the first household, followed by the systematic exclusion of three households before approaching the next eligible one. One adult resident per household was invited to participate. Eligibility criteria included being a community member aged 18 years or older, residing in the peri-urban areas surrounding Dar es Salaam, and having no prior history of cancer. If an individual declined participation or did not meet the inclusion criteria, the subsequent household was approached.

The sample size for both cases and controls was calculated using Epi Info version 7. Assuming a 1:1 case-to-control ratio, a total sample size of 600 participants was calculated to provide 90% statistical power to detect an odds ratio of 1.5, with a 95% confidence interval.

### 2.3. Data Collection

Data was collected via structured interviews using a structured electronic questionnaire tool installed in the Kobo toolbox that captured established HNC risk factors. The first section of the interview gathered information on socio-demographic details, such as the participant’s birthplace and place of residence. It also included questions about the participant’s family history of cancer. A detailed history of tobacco smoking and alcohol use was obtained, capturing specific information such as use, age at initiation, cessation, and the frequency and quantity consumed.

Questions were asked about the frequency of food consumption for categories such as rice, bread, chips, beans, fresh and cooked vegetables, pickled vegetables, fruits, smoked fish and meat, boiled meat, pepper, Ugali, cassava, ground nuts, and salt-preserved food. The frequency of consumption was recorded as daily, 3–5 times per week, 1–2 times per week, less than 1 time per week, and at an unknown frequency. 

The full occupational history of each participant was collected, including a detailed description of each job held for more than six months since the age of 18. The occupational information collected included the start and termination dates of each occupation. The main occupations of participants’ mothers and fathers were also registered. Occupations were coded according to the International Standard Classification of Occupations (ISCO 08) and the Tanzania Standard Classification of Occupations (TASCO). The principal investigator, with assistance from trained research assistants, conducted the interviews in Kiswahili.

### 2.4. Statistical Analysis

Descriptive statistics were used to summarize socio-demographic factors and exposure variables for cases and controls. The arithmetic mean (AM) and standard deviation (SD) for age were calculated after the Kolmogorov–Smirnov test (D = 0.0000, *p*-value = 1.000) showed that age data were normally distributed. Collinearity testing was conducted to assess potential multicollinearity among the predictor variables. The high correlation of 0.86 obtained for age and working experience groups suggested significant collinearity. Therefore, one of these variables was omitted while running a regression model. 

A univariate logistic regression model was used to test the association between each instance of exposure and HNC risk. The odds ratio (OR) of HNCs and the 95% confidence interval (CI) were calculated for the variables behind each type of exposure (occupations, sex, age group, birthplace, smoking, and alcohol use). Occupational exposure was classified by the ISCO 08 into the following groups: 5 Service and Sales Workers, 5120 Cooks, 71 Building and Related Trades Workers, and 921 Agricultural Workers. For some analyses, these occupational groups were further categorized into two major groups: agriculture and non-agriculture workers. Multivariate logistic regression was used to estimate ORs and 95% CIs, adjusting for birthplace (rural, urban), ever smoked (yes, no), ever used alcohol (yes, no), and age group (<40, 40–60, >60). Age groups were based on data from the literature, where the risk age for cancer starts at 40. A regression analysis was also performed separately for participants who had been working for more than 10 years, as there is a latency between exposure and the development of cancer. Potential confounders were included in the model, and the difference between cases and controls was *p* > 0.10.

Smoking status was classified as ever smoked, including those who currently smoke and those who quit smoking, while current smokers included only those who smoked at the time of the survey. Alcohol use status was classified as having ever used alcohol, including those who currently use alcohol and those who quit, while current alcohol users included only those who still consumed alcohol at the time of the survey. Dietary factors were identified from a literature review and included smoked fish, smoked meat, and fried foods/potatoes. These dietary factors have been recognized as high-risk factors for HNCs [13,14,15] and were incorporated into the model in this study. 

To account for the potential confounding effects of smoking, the data were stratified according to smoking status. This approach allowed the relationship between other exposure variables to be examined while controlling for smoking. Further stratification was conducted based on both smoking status and alcohol consumption, allowing for the independent evaluation of exposure variables for both smoking and alcohol use. However, due to the small number of smoking and alcohol use groups, these results were not included in the tables (Tables 3 and 4). In addition, given the extremely low frequency of family cancer history, this variable was excluded from the models. All data analyses were performed using Stata/SE version 15.1, with a significance level of *p* less than 0.05.

## 3. Results

A total of 298 HNC cases and 305 controls were recruited in this study. All patients and controls, except one, participated in the study. The cases were older than the controls, with mean ages of 51 and 43 years, respectively. The sex distribution indicated a greater number of men for both cases and controls; however, the differences were borderline significant. Of the cases, 78.2% were born in rural areas, and a similar figure was found controls at 32.5%. A significant difference was also observed between cases and controls regarding current place of residence, ever smoked tobacco, current tobacco smoking, ever used alcohol, currently using alcohol, cooking, and eating smoked meat, smoked fish, and fried potatoes. The results showed no statistically significant differences between the cases and controls regarding education or family history of cancer. Agricultural occupation was registered for 51.7% of the cases compared to 22% of the controls (Table 1). 

This study also assessed the occupational history of both the father and mother for cases and controls; over 80% of cases had parents engaged in agriculture, while this figure was over 50% for the control group.

Multivariate logistic regression analysis for the independent associations of HNC risk factors indicated an elevated risk of HNC among participants with a history of agricultural work: OR = 3.7 (95% CI = 2.62–5.30). After adjusting for age, sex, birthplace, smoking, alcohol, and eating smoked meat, smoked fish, and fried potatoes, the adjusted odds ratio for agricultural work was OR = 2.6 (95% CI = 1.60–4.37), indicating that this might be a significant factor associated with HNC (Table 2). 

In a similar regression analysis of non-smokers only (*n* = 124), agricultural work was associated with an increased risk of HNC, with crude OR = 3.3 (95% CI = 2.12–4.98). After adjusting for age, sex, birthplace, smoking, alcohol, and eating smoked meat, smoked fish, and fried potatoes, the odds ratio was 2.6 (95% CI = 1.48–4.53), indicating that working in agriculture remained a significant risk factor for HNC (Table 3).

The analysis was also performed for non-smokers and non-alcohol users only (*n* = 363). In this group, working in agriculture was associated with an increased risk of HNC estimated at OR = 3.0 (95% CI = 1.88–4.66). After adjusting for age, sex, birthplace, alcohol, and eating smoked meat, smoked fish, and fried potatoes, the odds ratio for agricultural work was 3.0 (95% CI = 1.52–5.97), indicating that agricultural work remained a significant risk factor for HNC in both the non-smoking and non-alcohol use groups (Table 4).

As the control group was younger than the cases, and as cancer is known to have a latency period, the analysis was also performed among participants who had worked more than ten years in agriculture. Working in agriculture was associated with the increased risk of HNC, as shown by a crude OR = 7.2 and 95% CI of 4.46–11.58. After adjusting for age, sex, birthplace, smoking, alcohol use, and eating smoked meat, smoked fish, and fried potatoes, the adjusted odds ratio for agriculture workers was 5.1 (95% CI: 2.56–9.94) (Figure 1). A table showing the results for patients who worked less than 10 years and those who worked more than 10 years is included in the Appendix A.

## 4. Discussion

This study demonstrates a significant association between HNCs and employment in the agricultural sector. Notably, this relationship was evident even after excluding individuals with histories of smoking and alcohol consumption. The risk is particularly elevated among participants who have been working in this sector for more than ten years.

Our findings reveal that over 52% of HNC cases were engaged in agricultural activities. This is in line with previous studies. Agricultural work is frequently linked to pesticide exposure, which has been associated with an increased risk of HNCs in studies conducted in Iran [11]. Similar patterns have been observed in research from the United States, where more than 50% of HNC cases were identified in individuals working in agriculture [9,12]. This consistency is noteworthy despite contextual differences. For example, in regions such as the United States and Europe, less than 10% of the population is employed in agriculture [16] compared to Tanzania, where over 60% of the population is engaged in agricultural work [17]. Therefore, the finding that more than half of HNC cases in our study were identified in agricultural workers underscores the potential health risks associated with agricultural exposure and highlights the need for further investigation. It is highly warranted to monitor the use of pesticides and the exposure of agricultural workers in developing countries. 

Additionally, our results are consistent with a previous case–control study on pharyngeal cancer conducted in Spain, which reported an increased cancer risk associated with pesticide exposure, even after adjusting for smoking and alcohol consumption [18]. This elevated risk was attributed to exposure during farming activities.

Head and neck cancer patients show marked differences in age distribution between developing and developed countries. Evidence indicates that in developed regions, the majority of HNC patients have a mean age above 60 years [6,19,20]. In contrast, our study found a mean age of 51 years among HNC cases, with over 71% of patients being under the age of 60. This age distribution aligns with findings from other cancer studies conducted in Africa [21]. The earlier onset of HNC observed in developing countries may be attributed to multiple factors, including occupational exposure and suboptimal working conditions that contribute to increased health risks.

The role of smoking in the etiology of HNCs is well established. Numerous studies conducted across Africa, Europe, and the United States have consistently demonstrated an elevated risk of HNC among individuals with a history of tobacco use [12,19,20]. In our study, 28.2% of HNC cases reported a history of smoking, with only 4.8% identified as current smokers. In comparison, 13% of the control group had been smoking, and among those, more than 52% were current smokers. These findings contrast with previous studies conducted in Croatia and France, where the prevalence of smoking among HNC cases exceeded 80% [19,20]. The observed differences in smoking prevalence among cases likely reflect broader population-level trends. In Tanzania, the national smoking prevalence is approximately 8% [16], whereas in France and Croatia, it exceeds 30% [19,20]. This contextual difference underscores the importance of interpreting smoking-related cancer risk within the demographic and behavioral patterns of each population.

To evaluate the impact of occupation independent of smoking and alcohol use, we stratified our data based on participants’ smoking and alcohol consumption status. Our findings indicate that among individuals who neither smoked nor consumed alcohol, those with an occupation in agriculture exhibited a significantly elevated risk of developing HNCs. These results align with a previous case–control study conducted in Iran [11], which reported a strong association between agricultural work and HNC incidence after excluding smokers and alcohol users from the analysis.

Evidence shows that HNCs are more common in men than women. This evidence is consistently demonstrated in various studies from developed and developing countries. In this study, 56% of cases were men; however, this number is not significantly different from that of women. This finding differs from studies conducted in the Netherlands and Croatia, where more than three-quarters of cases were men [6,19]. The gender differences between cases likely reflect differences in exposure to risk factors such as smoking, alcohol use, and other risk factors that are more prevalent in men than women. However, this study’s findings show that Tanzania is different, with an almost equal number of cases among men and women, indicating that men and women have nearly similar exposure levels to risk factors.

A family history of cancer is a recognized risk factor for the development of various malignancies, including HNCs. In our study, 10.4% of HNC cases and 12.5% of controls reported a family history of cancer. These figures contrast sharply with findings from studies conducted in the United States and Croatia, where the reported prevalence of family history among cancer cases was 48% and 54%, respectively [17,19]. The observed discrepancies may be attributed to differences in healthcare infrastructure, diagnostic capabilities, and public awareness between developed and developing countries. In high-income settings, widespread access to healthcare services, robust diagnostic systems, and greater health literacy are likely to contribute to increased recognition and reporting of familial cancer history. In contrast, limited access to medical care and lower levels of awareness in developing regions may result in underreporting or a lack of knowledge regarding family history of cancer.

There is a clear relationship between the duration of agricultural work—or length of exposure—and the risk of developing HNCs. Our findings indicate an increased risk of HNC among participants who have been engaged in agricultural work for more than ten years. These results are consistent with previous case–control studies [4,20,22], which report similar results, where the risk of developing HNC rises significantly after ten or more years in this occupation.

This study has several limitations. Most of the data on risk factors were self-reported by the participants and may be subject to recall bias. However, interviewers dedicated time during data collection to ensure the information obtained was as accurate and reliable as possible. Additionally, our findings may be affected by selection bias due to geographical differences in residence between cases and controls. Nonetheless, this approach to control selection was preferred over recruiting patients or family members, as these groups may share underlying health conditions or risk factors with the cases, potentially confounding the observed associations. However, the controls were younger than the cases. This was controlled by adjustments for age in the statistical analyses and performing a separate analysis of participants with work duration above ten years. 

Despite these limitations, this study has notable strengths. It is among the first to investigate the association between occupation and HNC, both globally and within the Tanzanian context. Importantly, occupational data was based on the participants’ primary long-term employment, minimizing the risk of misclassification. Furthermore, the collection of detailed information on lifestyle factors, diet, family history of cancer, and geographical origin and residence allowed the adjustment of key confounding variables. Finally, the response rate almost reached 100%, highlighting the representativeness of the populations included.

## 5. Conclusions

Our findings indicate that employment in the agricultural sector is associated with an elevated risk of developing HNCs, even after controlling established risk factors such as smoking and alcohol consumption. Carcinogens in agriculture should be examined in future studies.

## Figures and Tables

**Figure 1 ijerph-22-01643-f001:**
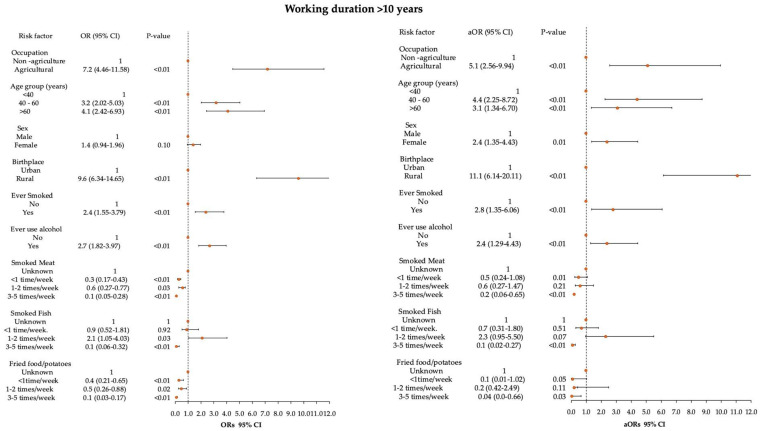
Forest plot showing the relationship between head and neck cancer and occupation, adjusted for other risk factors in a multivariate logistic regression, including patients with work experience >10 years. The dots indicate the point estimate, odds ratio (OR) or adjusted OR. The horizontal lines represent the 95% confidence interval.

**Table 1 ijerph-22-01643-t001:** Socio-demographic, lifestyle, diet and work factors among head and neck cancer cases and a community control group in Tanzania. Different HNCs are described at the end *(n* = 603).

Characteristics	Cases (*n* = 298)		Controls (*n* = 305)		*p*-Value ^1^
Age in Years Mean (SD)	51 (15.4)		43 (15.9)		
	*n*	%	*n*	%	
**Age group (years)**					
<40	72	24.2	152	49.8	<0.001
40–60	142	47.7	106	34.7	
>60	84	28.2	47	15.4	
**Sex**					
Female	130	43.6	109	35.7	0.05
Male	168	56.4	196	64.3	
**Educational level**					
Informal	51	17.1	42	13.8	0.50
Primary School	174	58.4	182	59.6	
Secondary and above	73	24.5	81	26.6	
**Marital Status**					
Unmarried	97	32.5	130	42.6	<0.001
Married	201	67.5	175	57.4	
**Birthplace**					
Urban	65	21.8	206	67.5	<0.001
Rural	233	78.2	99	32.5	
**Residence place**					
Urban	127	42.6	217	71.2	<0.001
Rural	171	57.4	88	28.9	
**Family history of cancer**					
No	267	89.6	267	87.5	0.42
Yes	31	10.4	38	12.5	
**Ever smoked tobacco**					
No	214	71.8	265	86.9	<0.001
Yes	84	28.2	40	13.1	
**Current tobacco smoker**					
No	80	94.2	19	47.5	<0.001
Yes	4	4.8	21	52.5	
**Ever used alcohol**					
No	168	56.4	231	75.7	<0.001
Yes	130	43.6	74	24.3	
**Currently using alcohol**					
No	123	94.6	31	41.9	<0.001
Yes	7	5.4	43	58.1	
**Cooking**					
Firewood and/or charcoal	276	92.6	263	86.2	<0.001
Gas	22	7.4	42	13.8	
**Occupations ***					
Service and Sales Workers	99	33.2	152	49.8	<0.001
Cooks	23	7.7	49	16.1	
Building and Related Trades	22	7.4	36	11.8	
Sum non-agriculture	144	48.3	237	77.7	
Agriculture	154	51.7	68	22.3	
**Working experience (years)**					
<10	34	11.4	71	23.3	<0.001
10–20	31	10.4	69	22.6	
21–30	61	20.5	50	16.4	
31–40	75	25.2	60	19.7	
>40	97	32.6	55	18.0	
**Smoked Meat**					
Unknown	99	33.3	39	14.7	<0.001
<1 time/week	110	36.9	130	49.1	
1–2 times/week	80	26.8	60	22.6	
3–5 times/week	9	3.0	36	13.6	
**Smoked Fish**					
Unknown	32	10.7	30	9.8	<0.001
<1 time/week	132	44.3	126	41.4	
1–2 times/week	121	40.6	52	17.0	
3–5 times/week	13	4.4	97	31.8	
**Chips/fried potatoes**					
Unknown	78	26.2	32	10.4	<0.001
<1 time/week	125	41.9	131	43.0	
1–2 times/week	82	27.5	70	23.0	
3–5 times/week	13	4.4	72	23.6	
**Head and neck cancer types**					
Pharynx	82	27.5		-	
Larynx	35	11.7	-	-	
Oral cavity	92	30.9	-	-	
Nasal Cavity and Paranasal Sinuses	89	29.9	-	-	

* Longest held job. Here, agriculture and the sum of other occupations are compared; ^1^ chi-square test. (“-” means no cancer present, as this is a table also for controls).

**Table 2 ijerph-22-01643-t002:** The association between occupation and head and neck cancer in a case–control study in Tanzania. The control group was a community population; multivariate logistic regression was used to adjust for age, sex, birthplace, smoking, alcohol, and eating smoked meat, smoked fish, and fried potatoes. The risk of adjusted factors is also presented for occupation and all the other factors.

Risk Factors	Cases *(n* = 298)	Control (*n* = 305)	OR	*p*-Value	aOR	*p*-Value
	*n* (%)	*n* (%)	(95% CI)		(95% CI)	
**Occupation**						
Non-agriculture	144 (48.3)	237 (77.7)	1		1	
Agricultural	154 (51.7)	68 (22.3)	3.7 (2.62–5.30)	<0.01	2.6 (1.60–4.37)	<0.01
**Age group (years)**						
<40	72 (24.2)	152 (49.8)	1		1	
40–60	142 (47.7)	106 (34.7)	2.8 (1.94–4.12)	<0.01	3.2 (1.86–5.39)	<0.01
>60	84 (28.2)	47 (15.4)	3.8 (2.40–5.94)	<0.01	2.2 (1.15–4.17)	<0.02
**Sex**						
Male	168 (56.4)	196 (64.3)	1			
Female	130 (43.6)	109 (35.7)	1.4 (1.00–1.93)	0.05	2.1 (1.27–3.43)	<0.01
**Birthplace**						
Urban	65 (21.8)	206 (67.5)	1			
Rural	233 (78.2)	99 (32.5)	7.5 (5.18–10.74)	<0.01	8.6 (5.23–14.06)	<0.01
**Ever smoked**						
No	214 (71.8)	265 (86.9)	1		1	
Yes	84 (28.2)	40 (13.1)	2.6 (1.71–3.94)	<0.01	3.0 (1.56–5.94)	<0.01
**Ever used alcohol**						
No	168 (56.4)	231 (75.7)	1		1	
Yes	130 (43.6)	74 (24.3)	2.4 (1.701–3.42)	<0.01	1.72 (1.94–2.94)	0.05
**Smoked Meat**						
Unknown	99 (33.2)	48 (15.7)	1			
<1 time/week	110 (36.9)	148 (48.5)	0.4 (0.23–0.55)	<0.01	0.4 (0.22–0.80)	0.01
1–2 times/week	80 (26.9)	68 (22.3)	0.6 (0.36–0.92)	0.02	0.6 (0.31–1.30)	0.21
3–5 times/week	9 (3.0)	41 (13.4)	0.1 (0.05–0.24)	<0.01	0.2 (0.06–0.50)	<0.01
**Smoked Fish**						
Unknown	32 (10.7)	30 (9.8)	1		1	
<1 time/week	132 (44.3)	126 (41.3)	1.0 (0.56–1.71)	0.95	1.1 (0.48–2.32)	0.9
1–2 times/week	121 (40.6)	52 (17.1)	2.2 (1.20–3.95)	0.01	2.7 (1.24–6.02)	0.01
3–5 times/week	13 (4.6)	97 (31.8)	0.1 (0.10–0.27)	0.00	0.1 (0.04–0.31)	<0.01
**Chips/fried potatoes**						
Unknown	78 (26.2)	32 (10.5)	1		1	
<1 time/week	125 (42.0)	131 (43.0)	0.3 (0.24–0.63)	<0.01	0.5 (0.26–103)	0.06
1–2 times/week	82 (27.5)	70 (23.0)	0.5 (0.3–0.81)	<0.01	0.8 (0.40–1.78)	0.66
3–5 times/week	13 (4.4)	72 (23.6)	0.1 (0.04–0.15)	<0.01	0.2 (0.07–0.48)	0.001

OR = odds ratio; CI = confidence interval; aOR = adjusted odds ratio.

**Table 3 ijerph-22-01643-t003:** The association between occupation and head and neck cancer among 479 non-smokers in a case–control study in Tanzania. The control group was a community population; multivariate logistic regression was used to adjust for age, sex, birthplace, alcohol, and eating smoked meat, smoked fish, and fried potatoes. The risk of adjustment factors is also presented for occupation and all the other factors.

		Non-Smoking Group					
Risk Factor		Cases *(n* = 214)	Control *(n* = 265)	OR (95% CI)	*p*-Value	aOR (95% CI)	*p*-Value
		*n* (%)	*n* (%)				
**Occupation**							
Non–agriculture		106 (49.5)	205 (77.4)	1	1	1	
Agriculture		108 (50.5)	60 (22.6)	3.3 (2.12–4.98)	<0.01	2.6 (1.48–4.53)	<0.01
**Age group (years)**							
<40		67 (31.3)	136 (51.3)	1		1	
40–60		97 (45.3)	88 (33.2)	2.2 (1.48–3.37)	<0.01	3.2 (1.80–5.90)	<0.01
>60		50 (23.4)	41 (15.5)	2.4 (1.49–4.11)	<0.01	1.9 (0.92–3.90)	0.08
**Sex**							
Male		121 (56.5)	108 (40.8)	1		1	
Female		93 (43.5)	157 (59.3)	1.9 (1.31–2.72)	<0.01	1.7 (1.12–2.92)	0.04
**Birthplace**							
Urban		43 (20.0)	180 (67.9)	1		1	
Rural		171 (80.0)	85 (32.1)	8.4 (5.52–12.84)	<0.01	11.0 (6.2115.56)	<0.01
**Smoked Meat**							
Unknown		81 (37.9)	39 (14.7)				
<1 time/week		72 (33.6)	130 (49.1)	0.3 (0.17–0.43)	<0.01	0.2 (0.11–0.52)	<0.01
1–2 times/week		57 (26.6)	60 (22.6)	0.5 (0.27–0.77)	0.004	0.5 (0.20–1.05)	0.07
3–5 times/week		4 (1.9)	36 (13.6)	0.1 (0.02-0.17)	<0.001	0.1 (0.02–0.32)	<0.01
**Smoked Fish**							
Unknown		25 (11.7)	25 (9.4)	1		1	
<1 time/week		90 (42.1)	110 (41.5)	0.8 (0.44–1.52)	0.53	1.4 (0.56–3.48)	0.45
1–2 times/week		92 (43.0)	46 (17.4)	2.0 (1.04–3.86)	0.04	4.0 (1.58–10.13)	<0.01
3–5 times/week		7 (3.3)	84 (31.7)	0.1 (0.03–0.22)	0.00	0.1 (0.04–0.42)	<0.01
**Chips/fried potatoes**							
Unknown		54 (25.2)	29 (10.9)	1		1	
<1 time/week		87 (40.7)	110 (41.5)	0.42 (0.24–0.72)	0.002	0.8 (0.35–1.78)	0.60
1–2 times/week		62 (29.0)	64 (24.2)	0.5 (0.29–0.92)	0.03	1.2 (0.49–1.05)	0.78
3–5 times/week		11 (5.1)	62 (23.4)	0.1 (0.04–0.20)	<0.01	0.4 (0.12–1.02)	0.05

OR = odds ratio; CI = confidence interval; aOR = adjusted odds ratio.

**Table 4 ijerph-22-01643-t004:** The association between occupation and head and neck cancer among 363 non-smokers and non-alcohol users in a case–control study in Tanzania. The control group was a community population; multivariate logistic regression was used to adjust for age, sex, birthplace, alcohol, and eating smoked meat, smoked fish, and fried potatoes. The risk of the adjustment factors is also presented for occupation and all the other factors.

	Non-Smokers and Non-Alcohol Users					
Risk Factor	Cases *(n* = 148)	Control (*n* = 215)	OR (95% CI)	*p*-Value	aOR (95% CI)	*p*-Value
	*n* (%)	*n* (%)				
**Occupation**						
Non-agriculture	80 (54.1)	167 (77.7)	1		1	
Agriculture	68 (46.0)	48 (22.3)	3.0 (1.88–4.66)	<0.01	3.0 (1.52–5.97)	<0.01
**Age group (years)**						
<40	54 (36.5)	112 (52.1)	1		1	
40–60	66 (44.6)	71 (33.0)	1.9 (1.21–3.07)	0.01	2.5 (1.25–4. 98)	<0.01
>60	28 (18.9)	32 (14.9)	1.8 (0.99–3.31)	0.05	1.2 (0.48–2.83)	0.74
**Sex**						
Male	60 (40.5)	85 (39.5)	1		1	
Female	88 (59.5)	130 (60.5)	2.2 (1.46–3.44)	<0.01	2.0 (1.063. 84)	0.03
**Birthplace**						
Urban	31 (21.0)	148 (68.8)	1		1	
Rural	117 (79.0)	67 (31.2)	8.3 (5.11–13.60)	<0.01	14 (7.24–28.19)	<0.01
**Smoked Meat**						
Unknown	58 (39.2)	34 (15.8)				
<1 time/week	52 (35.1)	106 (49.3)	0.3 (0.17–0.49)	<0.01	0.4 (0.15–0.98)	0.03
1–2 times/week	37 (25.0)	49 (22.8)	0.4 (0.24–0.80)	<0.01	0.6 (0.22–1.53)	0.27
3–5 times/week	1 (0.7)	26 (12.1)	0.02 (0.01–0.17)	<0.01	0.01 (0.00–0.31)	0.003
**Smoked Fish**						
Unknown	21 (14.2)	22 (10.2)				
<1 time/week	64 (43.2)	96 (44.7)	0.7 (0.36–1.37)	0.30	1.2 (0.42–3.56)	0.7
1–2 times/week	60 (40.6)	34 (15.8)	1.8 (0.89–3.84)	0.10	4.4 (1.51–13.13)	0.007
3–5 times/week	3 (2.0)	63 (29.3)	0.04 (0.01–0.18)	0.00	0.1 (0.01–0.26)	<0.001
**Chips/fried potatoes**						
Unknown	39 (26.4)	26 (12.1)				
<1 time/week	59 (39.9)	88 (40.9)	0.4 (0.24–0.81)	0.01	0.7 (0.42–3.56)	0.40
1–2 times/week	43 (29.1)	52 (24.2)	0.6 (0.29–1.05)	0.07	0.9 (0.31–2.39)	0.27
3–5 times/week	7 (4.7)	49 (22.8)	0.1 (0.03–0.24)	<0.01	0.3 (0.08–1.04)	0.06

## Data Availability

Data for this study that supports its findings are available upon reasonable request from the corresponding author.

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
