# Peer review of "Occupations and Risk of Head and Neck Cancers: A Case–Control Study in Tanzania"

_ijerph, 2025, doi:10.3390/ijerph22111643_

Round 1

Reviewer 1 Report

Comments and Suggestions for Authors

Dear authors, I congratulate you on preparing and submitting the manuscript.
Here are some general considerations about the work:
Lines 55 and 56: The authors state that working conditions in Iran are very different from those in developing countries, but they do not provide further explanations.
Line 87: Why were the controls selected if they were 18 or older, considering that the study focuses on a chronic degenerative disease? Didn't this criterion pose the risk of working with very young subjects?
Line 90: I believe the random sampling process for addresses in Dar es Salaam should be more clarified and detailed. I consider it very vague and superficial. Furthermore, I did not observe the presentation of a criterion for defining the sample size of cases and controls, which I consider a crucial issue.
Lines 101 to 103: Did the authors conduct interviews or questionnaires? I believe this was the second instrument. The titles of Tables 2, 3, and 4 should be revised and rewritten.
The conclusions return to the study's limitations, a classic feature of case-control studies. I don't think it would be appropriate to address them in the conclusions.

Finally, I noted that the study was not submitted to a research ethics committee.

Reviewer 2 Report

Comments and Suggestions for Authors

General comment

This is an important study on a pertinent issue in public health.  Cancer of the head and neck are said to be on the increase and an association with work in agriculture will unravel some of the causes of this dreaded disease.  I would like to see the authors make some exploration on the discussion of individual pesticides, mentioned by name, that have been toxicologically linked with these cancers.  In other words, they should attempt to give some mechanistic linkages between the pesticides and the cancers

Specific comments

  • Lines 37-38: The prevalence of 15.4% for head and neck cancers is too high.  Actually, the paper cited there is reporting prevalence of recurrence and not prevalence of occurrence.  Please check
  • Lines 42-43: “Cancer has multifactorial causes, which can be categorized into modifiable and non-modifiable risk factors”.  This sentence appears to equate causative factors to risk factors
  • Lines 50-70: Oral sex is increasingly being reported as a factor in throat and neck cancer.  I wonder if not exploring in your study, at least in the introduction.  Please see Farsi et al. 2015. Sexual behaviours and head and neck cancer: A systematic review and meta-analysis. Cancer epidemiology, 39(6), pp.1036-1046.

Reviewer 3 Report

Comments and Suggestions for Authors

The manuscript assesses the risk of head and neck cancer in Tasmanian citizens, accounting for different sociodemographic, lifestyle, and occupational factors. The manuscript is well organized and structured. Before recommending for publication, I suggest some minor revisions:

Please ensure all abbreviations are defined in the abstract and in its first citation in the text (e.g., HNCs).

In the introduction, the authors mentioned that “evidence suggests that industrial and agricultural workplaces in these regions (Africa?) often involve high exposure to carcinogens” (lines 58-59) – some examples of carcinogens could be given, contributing to elucidate on the hazardous carcinogenic substances/products. Also, when focusing on Tanzania and the use of carcinogenic chemicals in agriculture, the identification of main products/agents frequently used (authorized/not authorized) would be very valuable.

Were sociodemographic data (e.g., age, gender/sex, weight, height, etc.) collected in the questionnaire? Also, was environmental exposure included in the questionnaire? This information should also be mentioned in the experimental section.

Please check for some grammatical errors and typos (e.g., lines 144-145, 150, etc.).

The quality of Figure 1 should be improved.

The information given in lines 262-264 could be used to improve the discussion based on the main agriculture products and agents in the monitored areas.  

Round 2

Reviewer 1 Report

Comments and Suggestions for Authors

Dear authors, I have verified that the questions presented in the first stage of the review have been sufficiently answered. I consider myself satisfied with them.
I would like to take this opportunity to congratulate you again on your research.